# Model-free Safe Control
# for Zero-Violation Reinforcement Learning

**Weiye Zhao**     **Tairan He**     **Changliu Liu**
Carnegie Mellon University
{weiyezha, tairanh, cliu6}@andrew.cmu.edu

**Abstract:** While deep reinforcement learning (DRL) has impressive performance in a variety of continuous control tasks, one critical hurdle that limits the application of DRL to physical world is the lack of safety guarantees. It is challenging for DRL agents to persistently satisfy a hard state constraint (known as the safety specification) during training. On the other hand, safe control methods with safety guarantees have been extensively studied. However, to synthesize safe control, these methods require explicit analytical models of the dynamic system; but these models are usually not available in DRL. This paper presents a model-free safe control strategy to synthesize safeguards for DRL agents, which will ensure zero safety violation during training. In particular, we present an implicit safe set algorithm, which synthesizes the safety index (also called the barrier certificate) and the subsequent safe control law only by querying a black-box dynamic function (e.g., a digital twin simulator). The theoretical results indicate the implicit safe set algorithm guarantees *forward invariance* to the safe set. We validate the proposed method on the state-of-the-art safety benchmark Safety Gym. Results show that the proposed method achieves zero safety violation and gains $95\% \pm 9\%$ cumulative reward compared to state-of-the-art safe DRL methods. Moreover, the proposed method can easily scale to high-dimensional systems.

**Keywords:** Mobile Robots, Safe Reinforcement Learning, Safe Control

## 1   Introduction

Safety in terms of persistently satisfying a hard state constraint is always an important consideration in robotics. For example, vehicles should not crash into pedestrians; robot arms should not hit walls. More safety concerns are raised during robot learning, where the robot needs to figure out the optimal control policy by exploring the surrounding environment. It is not enough to merely penalize the robot for entering the unsafe states using *posterior* measures [1, 2]. The robot's action should indeed be constrained to not enter the unsafe states using *prior* measures.

Safe control methods that constrain the robot motion to persistently satisfy a hard safety constraint in predictable environments have been extensively studied. The most widely used ones are *energy function-based methods*. These methods [3, 4, 5, 6] first design an energy function such that the safe states are with low energy and then synthesize control laws that are constrained to make the system dissipate energy. Then the system will never leave the safe set (i.e., *forward invariance*). These methods were first introduced for deterministic systems. They have been recently extended to stochastic and uncertain systems [7, 8] (uncertainties in either the state measurements or the future evolution of the ego robot or obstacles) and combined with robot learning for safety assurance [9]. However, the major limitation of these methods is that they require *white-box analytical models* of the system dynamics (e.g., Kinematic Bicycle Model [10]) and/or the associated uncertainties, which are challenging to obtain during robot learning.

This paper proposes a model-free safe control method that enables the application of *energy function-based methods* to ensure robot safety in environments where no white-box analytical model is available. The model-free safe control is able to safeguard any robot learning algorithm. The key insight we have is that the safe control laws can be synthesized by only querying black-box non-analytical dynamic functions. These black-box functions can be high-fidelity digital twin simulators

5th Conference on Robot Learning (CoRL 2021), London, UK.

or deep neural network models, which are usually available or can be constructed by data-driven learning. The key questions that we want to answer are: 1) how to synthesize the energy function (in our case called the safety index) so that there always exists a feasible control input to dissipate energy at any state, when the dynamics are black-box; and 2) how to efficiently sample the control space to find the best safe control (e.g., the control that dissipates energy and optimizes the reward) with black-box dynamics. We propose a safety index design rule to address the first question, and a sample-efficient algorithm to perform black-box constrained optimization to address the second question. By combining these approaches with the (model-based) safe set algorithm [5], we propose the (model-free) implicit safe set algorithm (ISSA). We then use ISSA to safeguard deep reinforcement learning (DRL) agents. Extension to other *energy function-based methods* and robot learning approaches is straight forward and will be explored in the future.

The key contributions of this paper are summarized below:

- We propose two techniques to synthesize safe control laws without white-box models: a safety index design rule for mobile robots in the 2D plane and a sample-efficient black-box optimization algorithm using adaptive momemtum boundary approximation (AdamBA).
- We propose the implicit safe set algorithm using these two techniques, which guarantees to generate safe controls for all system states without knowing the explicit system dynamics. We show that ISSA can safeguard DRL agents to ensure zero safety violation during training in Safety Gym. Our code is available on Github.[1]

## 2 Problem Formulation

**Dynamics**   Let $x_t \in \mathcal{X} \subset \mathbb{R}^{n_x}$ be the robot state at time step $t$, where $n_x$ is the dimension of the state space $\mathcal{X}$; $u_t \in \mathcal{U} \subset \mathbb{R}^{n_u}$ be the control input to the robot at time step $t$, where $n_u$ is the dimension of the control space $\mathcal{U}$. The system dynamics are defined as:

$$x_{t+1} = f(x_t, u_t), \tag{1}$$

where $f : \mathcal{X} \times \mathcal{U} \to \mathcal{X}$ is a function that maps the current robot state and control to the robot state in the next time step. For simplicity, this paper considers deterministic dynamics. The proposed method can be easily extended to stochastic case through robust safe control [11, 12], which will be left for future work. Moreover, it is assumed that we can only access an implicit black-box form of $f$, e.g., as an implicit digital twin simulator or a deep neural network model. Note that the word *implicit* refers to that we evaluate $f(x, u)$ without any explicit knowledge or analytical form of $f(x, u)$.

**Safety Specification**   The safety specification requires that the system state should be constrained in a closed subset in the state space, called the safe set $\mathcal{X}_S$. The safe set can be represented by the zero-sublevel set of a continuous and piecewise smooth function $\phi_0 : \mathbb{R}^{n_x} \to \mathbb{R}$, i.e., $\mathcal{X}_S = \{x \mid \phi_0(x) \leq 0\}$. $\mathcal{X}_S$ and $\phi_0$ are directly specified by users. The design of $\phi_0$ is straightforward in most scenarios. For example, for collision avoidance, $\phi_0$ can be designed as the negative closest distance between the robot and environmental obstacles.

**Reward and Nominal Control**   A robot learning controller generates the nominal control which is subject to modification by the safeguard. The learning controller aims to maximize rewards in an infinite-horizon deterministic Markov decision process (MDP). An MDP is specified by a tuple $(\mathcal{X}, \mathcal{U}, \gamma, r, f)$, where $r : \mathcal{X} \times \mathcal{U} \to \mathbb{R}$ is the reward function, $0 \leq \gamma < 1$ is the discount factor, and $f$ is the deterministic system dynamics defined in (1).

**Problem**   Building on top of the nominal learning agent, the core problem of this paper is to synthesize a safeguard for the learning agent, which monitors and modifies the nominal control to ensure **forward invariance** in a subset of the safe set $\mathcal{X}_S$. *Forward invariance* of a set means that the robot state will never leave the set if it starts from the set. The reason why we need to find a subset instead of directly enforcing *forward invariance* of $\mathcal{X}_S$ is that $\mathcal{X}_S$ may contain states that will inevitably go to the unsafe set no matter what the control input is. These states need to be penalized when

---

[1] https://github.com/intelligent-control-lab/ISSA_CoRL21

we synthesize the energy function. For example, when a vehicle is moving toward an obstacle with high speed, it would be too late to stop. Even if the vehicle is safe now (if $\mathcal{X}_S$ only constrains the position), it will eventually collide with the obstacle (unsafe). Then we need to assign high energy values to these inevitably-unsafe states.

## 3 Related Work

**Safe Reinforcement Learning** Safe RL either considers soft safety constraints or hard safety constraints. Typical safe RL methods for soft safety constraints include risk-sensitive safe RL [13], Lagrangian methods [14] and constrained policy optimization (CPO) [15]. These methods are able to find policies that satisfy the safety constraint in expectation, but cannot ensure all visited states are safe. The methods that are more closely related to ours are safe RL methods with hard safety constraints. Richard *et al.* [7] proposes a general safe RL framework, which combines CBF-based controllers with RL algorithms to guarantee safety and guide the learning process by constraining the set of explorable policies. However, the theory guarantee of CBF-based controller strongly relies on a known control affine dynamics system, which restricts its application to general nonlinear dynamics systems. More related works of safe RL methods with hard safe constraints are summarized in Appendix A.

**Safe Control** Representative *energy function-based methods* for safe control include potential field methods [3], control barrier functions (CBF) [4], safe set algorithms (SSA) [5], sliding mode algorithms [6], and a wide variety of bio-inspired algorithms [16]. The first step to synthesize a safe controller is to compute a desired energy function offline such that 1) the low energy states are safe and 2) there always exists a feasible control input to dissipate the energy. SSA has introduced a rule-based approach to synthesize the energy function as a continuous, piece-wise smooth scalar function on the system state space $\phi : \mathbb{R}^{n_x} \rightarrow \mathbb{R}$. And the energy function $\phi(x)$ is called a safety index in this approach. The general form of the safety index was proposed as $\phi = \phi_0^* + k_1 \dot{\phi}_0 + \cdots + k_n \phi_0^{(n)}$ where 1) the roots of $1 + k_1 s + \ldots + k_n s^n = 0$ are all negative real (to ensure zero-overshooting of the original safety constraints); 2) the relative degree from $\phi_0^{(n)}$ to $u$ is one (to avoid singularity); and 3) $\phi_0^*$ defines the same set as $\phi_0$ (to nonlinear shape the gradient of $\phi$ at the boundary of the safe set). It is shown in [5] that if the control input is unbounded ($\mathcal{U} = \mathbb{R}^{n_u}$), then there always exist a control input that satisfies the constraint $\dot{\phi} \leq 0$ when $\phi = 0$; and if the control input always satisfies that constraint, then the set $\{x \mid \phi(x) \leq 0\} \cap \{x \mid \phi_0(x) \leq 0\}$ is forward invariant. In practice, the actual control signal is computed through a quadratic projection of the nominal control $u^r$ to the control constraint

$$u = \arg\min_{u \in \mathcal{U}} \|u - u^r\|^2 \text{ s.t. } \dot{\phi} \leq -\eta(\phi), \tag{2}$$

where $\dot{\phi} \leq -\eta(\phi)$ is a general form of the constraint; $\eta : \mathbb{R} \rightarrow \mathbb{R}$ is a non decreasing function that $\eta(0) \geq 0$. For example, in CBF, $\eta(\phi)$ is designed to be $\lambda\phi$ for some positive scalar $\lambda$. In SSA, $\eta(\phi)$ is designed to be a positive constant when $\phi \geq 0$ and $-\infty$ when $\phi < 0$. Note there are two major differences between this paper and the existing results. First, this paper considers constrained control space, which then require careful selection of the parameters in $\phi$ to make sure the control constraint $\mathcal{U}_S(x) := \{u \in \mathcal{U} \mid \dot{\phi} \leq -\eta(\phi)\}$ is nonempty for states that $\phi \geq 0$. Secondly and most importantly, this paper considers general black-box dynamics, while the existing work considers analytical control-affine dynamics. For analytical control-affine dynamic models, $\mathcal{U}_S(x)$ is essentially a half-space and the projection (2) can be efficiently computed by calling a quadratic programming solver. However, for black-box dynamics, this constraint is challenging to quantify.

## 4 Method

This section introduces the implicit safe set algorithm (ISSA), which is able to leverage *energy function-based methods* (SSA in particular) with black-box dynamics, and be used to safeguard any nominal policy. ISSA contains two parts: 1) a safety index synthesis rule to make sure $\mathcal{U}_S(x)$ is nonempty for all $x$, and 2) a sample-efficient black-box optimization method to solve the projection of the nominal control to $\mathcal{U}_S(x)$. With these two components, the overall pipeline of the implicit safe set algorithm is summarized as follows:

- **Offline:** Design the safety index $\phi(x)$ according to the safety index design rule.
- **Online:** Project nominal control into $\mathcal{U}_S(x)$ during online robot maneuvers.

For simplicity, we consider discrete-time control only since real-world robot control systems are always implemented in discrete-time. Thus, we consider the discrete-time set of safe control $\mathcal{U}_S^D(x) := \{u \in \mathcal{U} \mid \phi(f(x,u)) \leq \max\{\phi(x) - \eta, 0\}\}$ in the following discussion.

## 4.1 Synthesize Safety Index

In this paper, the safety constraint that we are specifically interested in is collision avoidance for mobile robots in 2D planes. We treat the robot and the obstacles as point-mass circles with bounded collision radius. The safety specification is define as $\phi_0 = \max_i \phi_{0_i}$, and $\phi_{0_i} = d_{min} - d_i$ where $d_i$ denotes the distance between the center points of the robot and the $i$-th obstacle (static or non-static). It is assumed that the dynamics $f(x, u)$ is a bounded Lipschitz function. Denote $a$ and $w$ as the relative acceleration and relative angular velocity of the robot in the obstacle frame, respectively, as shown in fig. 1. The safety index for collision avoidance in 2D will be synthesized without referring to the specific dynamic model, but under the following assumptions.

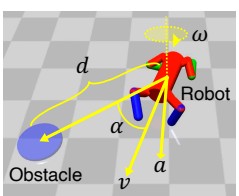

Figure 1: Notations.

**Assumption 1** (2D Collision Avoidance). *1) The state space is bounded, and the relative acceleration and angular velocity are bounded and both can achieve zeros, i.e., $w \in [w_{min}, w_{max}]$ for $w_{min} \leq 0 \leq w_{max}$ and $a \in [a_{min}, a_{max}]$ for $a_{min} \leq 0 \leq a_{max}$; 2) For all possible values of $a$ and $w$, there always exists a control $u$ to realize such $a$ and $w$; 3) The discrete-time system time step $dt \to 0$. 4) At any given time, there can at most be one obstacle becoming safety critical, such that $\phi - \eta \geq 0$ (Sparse Obstacle Environment).*

These assumptions are easy to met in practice. The bounds in the first assumption will be directly used to synthesize $\phi$. The second assumption enables us to turn the question on whether these exists a feasible control in $\mathcal{U}_S^D$ to the question on whether there exists $a$ and $w$ to decrease $\phi$. The third assumption ensures that the discrete time approximation error is small. The last assumption enables safety index design rule applicable with multiple moving obstacles. Following the rules in [5], we parameterize the safety index as $\phi = \max_i \phi_i$, and $\phi_i = \sigma + d_{min}^n - d_i^n - k\dot{d}_i$, where all $\phi_i$ share the same set of tunable parameters $\sigma, n, k, \eta \in \mathbb{R}^+$. Our goal is to choose these parameters such that $\mathcal{U}_S^D(x)$ is always nonempty. Under the above assumptions, the safety index design rule is constructed below.

**Safety Index Design Rule:** By setting $\eta = 0$, the parameters $k, n$, and $\sigma$ should be chosen such that

$$\frac{n(\sigma + d_{min}^n + kv_{max})^{\frac{n-1}{n}}}{k} \leq \frac{-a_{min}}{v_{max}}, \tag{3}$$

where $v_{max}$ is the maximum relative velocity that the robot can achieve in the obstacle frame.

## 4.2 Sample-Efficient Black-Box Constrained Optimization

The nominal control $u_t^r$ needs to be projected to $\mathcal{U}_S^D(x)$ by solving the following optimization:

$$\min_{u_t \in \mathcal{U}} \|u_t - u_t^r\|^2$$
$$\text{s.t. } \phi(f(x_t, u_t)) \leq \max\{\phi(x_t) - \eta, 0\} \tag{4}$$

Since the objective of (4) is convex, its optimal solution will always lie on the boundary of $\mathcal{U}_S^D(x)$ if $u^r \notin \mathcal{U}_S^D(x)$. Therefore, it is desired to have an efficient algorithm to find the safe controls on the boundary of $\mathcal{U}_S^D(x)$. To efficiently perform this black-box optimization, we propose a sample-efficient boundary approximation algorithm called Adaptive Momentum Boundary Approximation Algorithm (AdamBA), which is summarized in Algorithm 1 of Appendix B. We illustrate the main boundary approximation procedures of AdamBA in Figure 2, where AdamBA is supposed to find the boundary points of $\mathcal{U}_S^D(x)$ (green area) with respect to the reference control $u^r \notin \mathcal{U}_S^D(x)$ (red star). The core idea of the AdamBA follows the adaptive line search [17], where three main procedures are included. I. AdamBA first initialize several unit gradient vectors (green vectors) to be the sampling

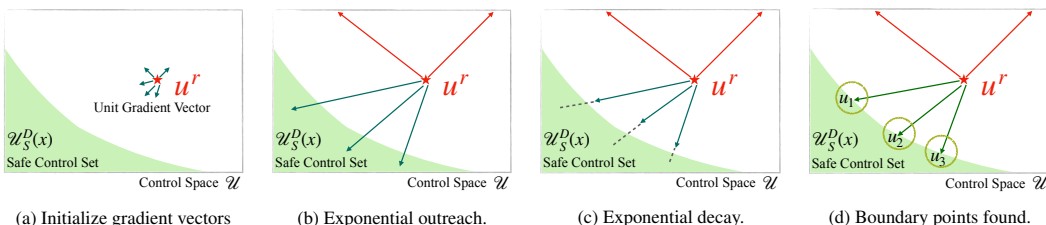

Figure 2: Illustration of the procedure of the AdamBA algorithm.

directions as shown in Figure 2a. II. AdamBA enters the *exponential outreach* phase by exponential increasing the gradient vector length until they reach $\mathcal{U}_S^D(x)$ as shown in Figure 2b. Note that we discard those gradient vectors that go out of control space (red vectors). III. Next, AdamBA enters *exponential decay* phase by iteratively applying binary search to find boundary points as shown in Figure 2c. Finally, a set of boundary points will be returned after AdamBA converges as shown in Figure 2d. Note that AdamBA and the line search methods are fundamentally similar to each other, while the purpose of AdamBA is to find the boundary of safe/unsafe action, while the line search methods are to find the minimum of a function. It can be shown that the boundary approximation error can be upper bounded, where the bound depends on the resolution of the sampling directions.

### 4.3 Implicit Safe Set Algorithm

Leveraging AdamBA and the safety index design rule, we construct the implicit safe set algorithm (ISSA). The proposed ISSA is summarized in Algorithm 2 of Appendix B. ISSA contains an offline stage and an online stage. In the offline stage, we synthesize the safety index according to the design rule (3). There are two major phases in the online stage for solving (4). In online-phase 1, we directly use AdamBA to find the safe controls on the boundary of $\mathcal{U}_S^D(x)$, and choose the control with minimum deviation from the reference control as the final output. In the case that no safe control is returned in online-phase 1 due to sparse sampling, online-phase 2 is activated. We uniformly sample the control space and deploy AdamBA again on these samples to find the safe control on the boundary of $\mathcal{U}_S^D(x)$. It can be shown that the ISSA algorithm is guaranteed to find a feasible solution of (4) and that solution ensures forward invariance to the set $\mathcal{S} := \mathcal{X}_S \cap \{x \mid \phi(x) \leq 0\}$.

Although the ISSA algorithm builds upon the safe set algorithm [5], the proposed safety index synthesis and AdamBA algorithm can be applied to other *energy function-based methods* to generate safe controls with or without an explicit analytical dynamics model.

## 5 Theoretical Results

We first present two propositions for the feasibility of the safety index synthesis rule and of Algorithm 2. The proofs for Proposition 1 and Proposition 2 are summarized in Appendix C and Appendix D, respectively. Then, we present the main theorem stating that the implicit safe set algorithm ensures forward invariance. The proof of Theorem 1 is summarized in Appendix E.

**Proposition 1** (Nonempty set of safe control). *If the dynamic system satisfies the assumptions in Assumption 1, then the safety index design rule in Section 4.1 ensures that the robot system in 2D plane has nonempty set of safe control at any state, i.e., $\mathcal{U}_S^D(x) \neq \emptyset, \forall x$.*

**Proposition 2** (Feasibility of Algorithm 2). *If the set of safe control is non-empty, Algorithm 2 can always find a local optimal solution of (4) with finite number of iterations.*

**Theorem 1** (Forward Invariance). *If the control system satisfies the assumptions in Assumption 1 and the safety index design follows the rule described in Section 4.1, the implicit safe set algorithm guarantees the forward invariance to the set $\mathcal{S} \subseteq \mathcal{X}_S$.*

## 6 Experimental Results

In our experiments, we aim to answer the following questions:

**Q1**: How does ISSA compare with other state-of-the-art methods for safe RL? Can ISSA achieve zero-violation of the safety constraint?

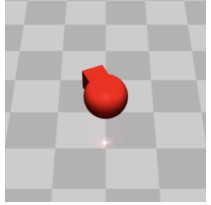 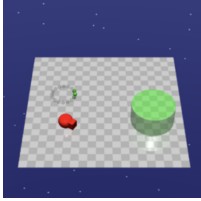 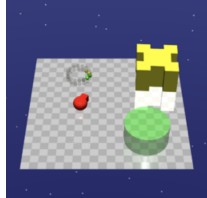 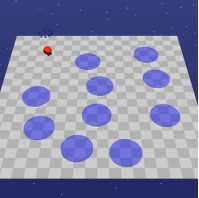 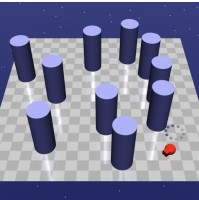

(a) Point robot: a simple 2D robot that can turn and move.

(b) Goal: navigating the robot inside the green goal area.

(c) Push: pushing the yellow box inside the green goal area.

(d) Hazards: non-physical dangerous areas.

(e) Pillars: fixed dangerous obstacles

Figure 3: The environmental settings for benchmark problems in Safety Gym.

**Q2**: How does the design of the safety index affect the set of safe control?

**Q3**: How do the hyper-parameters of ISSA and the dimensionality of the system impact its performance?

To demonstrate the effectiveness of the proposed implicit safe set algorithms, we conduct evaluation experiments on the safe reinforcement learning benchmark environment Safety Gym [14]. Our experiments adopt the Point robot ($\mathcal{U} \subseteq \mathbb{R}^2$) as shown in Figure 3a and the Doggo robot ($\mathcal{U} \subseteq \mathbb{R}^{12}$) as shown in Figure 1. We design 8 experimental environments with different task types, constraint types, constraint numbers and constraint sizes. We name these environments as `{Task}-{Constraint Type}{Constraint Number}-{Constraint Size}`. Note that `Constraint Size` equals $d_{min}$ in the safety index design. Two tasks are considered:

- `Goal`: The robot must navigate to a goal as shown in Figure 3b.
- `Push`: The robot must push a box to a goal as shown in Figure 3c.

And two different types of constraints are considered:

- `Hazard`: Dangerous (but admissible) areas as shown in Figure 3d. Hazards are circles on the ground. The agent is penalized for entering them.
- `Pillar`: Fixed obstacles as shown in Figure 3e. The agent is penalized for hitting them.

The methods in the comparison group include: unconstrained RL algorithm PPO [18] and constrained safe RL algorithms PPO-Lagrangian, CPO [15] and PPO-SL (PPO-Safety Layer) [19]. We select PPO as our baseline method since it is state-of-the-art and already has safety-constrained derivatives that can be tested off-the-shelf. We set the limit of cost to 0 for both PPO-Lagrangian and

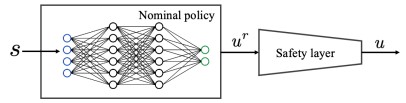

Figure 4: The PPO-ISSA structure.

CPO since we aim to avoid any violation of the constraints. To make sure ISSA can complete tasks while guaranteeing safety, we use a PPO agent as the nominal policy and ISSA as a safety layer to solve (4), we call this structure as PPO-ISSA, and it is illustrated in Figure 4. Such safety layer structure has also been used in PPO-SL [19] which leverages offline dataset to learn a linear safety-signal model and then construct a safety layer via analytical optimization. For all experiments, we use neural network policies with separate feedforward MLP policy and value networks of size (256, 256) with tanh activations. More details are provided in Appendix F.1.

## 6.1 Evaluating PPO-ISSA and Comparison Analysis

To compare the reward and safety performance of PPO-ISSA to the baseline methods in different tasks, constraint types, and constraint sizes, we design 4 test suites with 4 constraints which are summarized in Figure 6. The comparison results reported in Figure 6 demonstrate that PPO-ISSA is able to achieve zero average episode cost and zero cost rate across all experiments while slightly sacrificing the reward performance. The baseline soft safe RL methods (PPO-Lagrangian and CPO) fail to achieve zero-violation safety even when the cost limit is set to be 0. PPO-Lagrangian and CPO

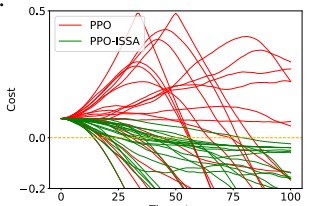

Figure 5: Cost changes over 100 time steps of PPO and PPO-ISSA starting from the same unsafe state over 20 trials.

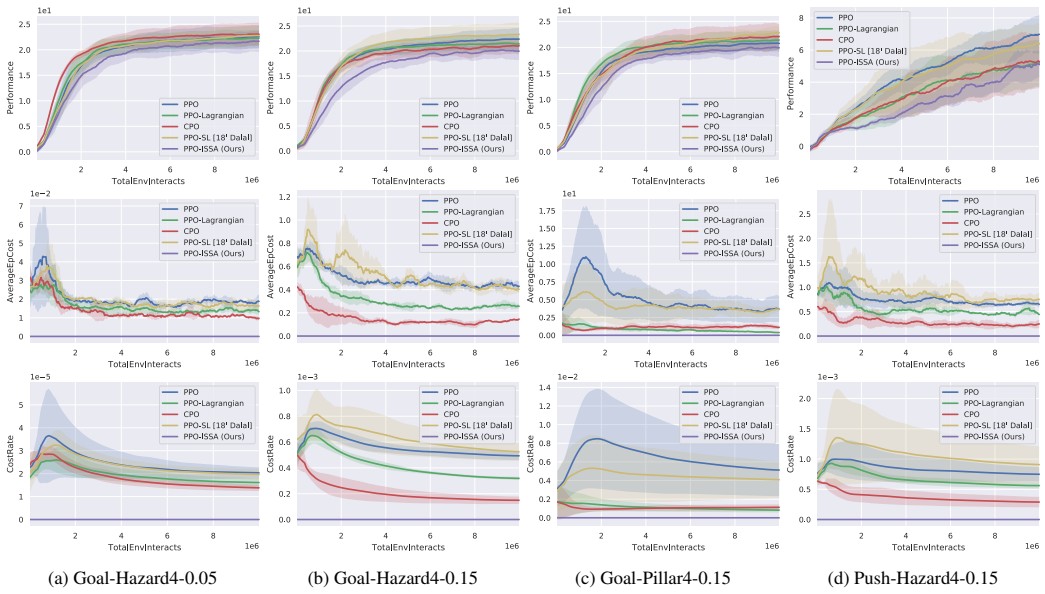

|     |     |     |     |
| --- | --- | --- | --- |
| (a) Goal-Hazard4-0.05 | (b) Goal-Hazard4-0.15 | (c) Goal-Pillar4-0.15 | (d) Push-Hazard4-0.15 |

Figure 6: Average episodic return, episodic cost and overall cost rate of constraints of PPO-ISSA and baseline methods on 4-constraint environments over five seeds.

fail since both methods rely on trial-and-error to enforce constraints while ISSA is able to guarantee forward invariance by Theorem 1. We also observe that PPO-SL fails to lower the violation during training, due to the fact that the linear approximation of cost function $c(x_{t+1}) \approx c(x_t) + g(x_t, w)^T u$ [19] becomes inaccurate when the dynamics are highly nonlinear like the ones we used in MuJoCo [20]. More importantly, PPO-SL cannot guarantee that these always exist a feasible safe control to lower the cost, since they directly use the user defined cost function which cannot always ensure feasibility. More detailed metrics for comparison and experimental results on test suites with 1 constraint are summarized in Appendix F.1.4.

We further compare the cost evolution of PPO and PPO-ISSA agents when starting from the same unsafe state (i.e., cost > 0). The comparison results are shown in Figure 5, which shows that PPO-ISSA can converge to $\mathcal{X}_S$ within 100 time steps across all experiments while cost evolution of PPO agents fluctuates wildly without preference to converge to safe set.

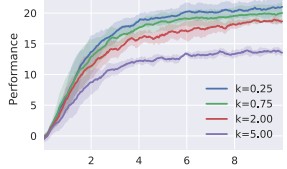

Figure 7: Average return of PPO-ISSA with different safety index design on Goal-Hazard4-0.15.

## 6.2 Feasibility of Safety Index Synthesis

To demonstrate how the set of safe control is impacted by different safety index definition, we randomly pick an unsafe state $x^*$ such that $\phi(x^*) > 0$, and visualize the corresponding set of safe control $\mathcal{U}_S^D$ under different safety index definitions, which are shown in Figure 8. Red area means $\Delta\phi > 0$ (i.e. unsafe control) and blue area means $\Delta\phi < 0$ (i.e. safe control). Figure 8a shows the set of safe control of distance safety index $\phi_d = \sigma + d_{min} - d$, which is the default cost definition of Safety Gym. The heatmap is all red in Figure 8a, which means that the set of safe control under the default $\phi_d$ is empty. Figure 8b shows the set of safe control of the synthesized safety index $\phi = \sigma + d_{min}^2 - d^2 - k\dot{d}$ with different value of $k$. With the synthesized safety index, Figure 8b demonstrates that the size of the set of safe control grows as $k$ increases, which aligns with the safety index synthesis rule discussed in Section 4.1 as larger $k$ is easier to satisfy (3). To demonstrate the reward performance of PPO-ISSA under different safety index designs, we select Goal-Hazard4-0.15 test suite. Figure 7 demonstrates the average return of PPO-ISSA under different value of $k$, which shows that the reward performance of PPO-ISSA deteriorates as $k$ value increases (since larger $k$ makes the control more conservative). Note that the set of safe control increases as the $k$ value increases, thus the optimal $k$ should be the smallest $k$ that makes the set of safe control nonempty for all states. Our safety index synthesis rule in (3) provides the condition to pick the optimal $k$.

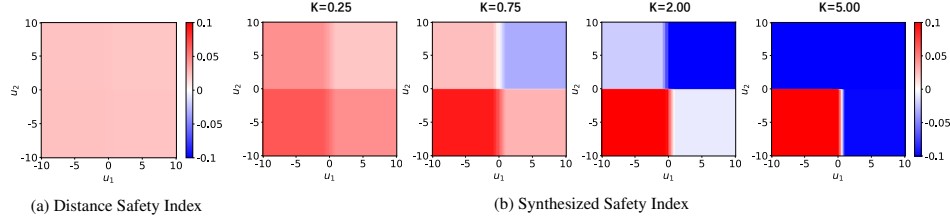

(a) Distance Safety Index                         (b) Synthesized Safety Index

Figure 8: Heat maps of the difference of safety index $\Delta\phi = \phi(f(x, u)) - \phi(x)$. The x-axis $u_1$ represents the control space of moving actuator, and the y-axis $u_2$ represents the control space of turning actuator.

| Number of vectors | Simulation Time $T_{sim}$ | Overall ISSA Time $T_{all}$ | Return $\bar{J}_r$ |
|---|---|---|---|
| $n = 3$ | 0.297 | 0.301 | 0.738 |
| $n = 5$ | 0.504 | 0.511 | 0.826 |
| $n = 10$ | 0.987 | 1.000 | 1.000 |

Table 1: Normalized computation time and return under different number of vectors in ISSA. These results are average on 100 ISSA runs over five random seeds on Goal-Hazard4-0.15.

## 6.3 Sensitivity Analysis and Scalability Analysis

To demonstrate the scalability and the performance of PPO-ISSA when ISSA chooses different parameters, we conduct additional tests using the test suite Goal-Hazard4-0.15. Among all input parameters of ISSA, the gradient vector number $n$ is critical to impact the quality of the solution of (4). Note in the limit when $n \to \infty$, ISSA is able to traverse all boundary points of the set of safe control, hence able to find the global optima of (4). We pick three different $n$ values: 3, 5, 10; and report the average episode reward of PPO-ISSA, and the computation time of ISSA when solving (4), which includes the normalized average ISSA computation time and the normalized average simulation time for each run. The results are summarized in Table 1, which demonstrates that the reward performance of PPO-ISSA would improve as $n$ gets bigger since we get better optima of Equation (4). In practice, we find that the reward performance will stop improving when $n$ is big enough ($n > 10$).

The computation time scales linearly with respect to $n$ while the majority (98%) of computation cost is used for environment simulation, which can be improved in the future by replacing the simulator with a more computationally efficient surrogate model. We also evaluate the scalability of ISSA in a 12-dimensional control system, showing ISSA is able to achieve zero-violation and best performance with acceptable computation cost in Figure 9. Besides, ISSA can be largely accelerated using parallel computation. The detailed discussion can be found in Appendix G.

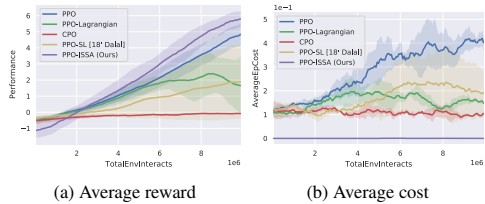

(a) Average reward      (b) Average cost

Figure 9: Average episodic return and episodic cost of PPO-ISSA and baseline methods on Goal-hazard1-0.15 environment of a doggo robot over five seeds.

## 7 Conclusion and Future Work

Safety guarantee is critical for robotic applications in real world, such that robots can persistently satisfy safety constraints. This paper presents a model-free safe control strategy to synthesize safeguards for DRL agents, which will ensure zero safety violation during training. In particular, we present an implicit safe set algorithm as a safeguard, which synthesizes the safety index (also called the barrier certificate) and the subsequent safe control law only by querying a black-box dynamics function (e.g., a digital twin simulator). The theoretical results indicate that the synthesized safety index guarantees nonempty set of safe control for all system states, and ISSA guarantees *forward invariance* to the safe set. We further validate the proposed safeguard with DRL on state-of-the-art safety benchmark Safety Gym. Our proposed method achieves zero safety violation and $95\% \pm 9\%$ reward performance compared to state-of-the-art safe DRL methods.

There are two major directions for future work. Firstly, we will further generalize the safety index synthesis rule to cover a wider range of applications other than collision avoidance in 2D. Secondly, we will further speed up the implicit model evaluation step by replacing the physical engine based simulator with a learned surrogate model while taking the learned dynamics error into account.

**Acknowledgments**

This work is supported by Amazon Research Award.

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
