# OpenReview forum: "Model-free Safe Control for Zero-Violation Reinforcement Learning"
_robot-learning.org/CoRL/2021/Conference — CoRL2021 Poster_

### Official Review · Reviewer_nDua · 2021-07-14

**Originality:** Fair
**Technical Quality:** Good
**Clarity Of Presentation:** Good
**Impact:** 2

**Recommendation:**

Weak Accept: I recommend accepting the paper, but will not argue for my recommendation if the majority of other reviewers have a different opinion.

**Summary:**

This paper proposed a method, Implicit Safe Set Algorithm (ISSA), which focuses on collision avoidance. ISSA contains two parts: a parameter design rule and a sample-based optimization method. Firstly, the original constraints are reformulated into the Safe Index Form [1] that includes hyperparameters. And the hyperparameters should satisfy the parameter design rule, which incorporates the maximum velocity and acceleration. Secondly, a sample-based line search method AdamBA is applied to find the closest control action w.r.t the nominal action from the RL agent. The proposed method is tested in the Safety Gym environment.

**Issues:**

1. There is no Figure 10a in the appendix.
2. Need more explanation about the safety index rule. It is not clear how the rule is determined.
3. “ It can be shown that the ISSA algorithm is guaranteed to find a feasible solution ...”. How?

[1] Liu, C., & Tomizuka, M. (2014). Control in a safe set: Addressing safety in human-robot interactions. ASME 2014 Dynamic Systems and Control Conference (DSCC), 3(November). https://doi.org/10.1115/DSCC2014-6048

[2] Dalal, G., Dvijotham, K., Vecerik, M., Hester, T., Paduraru, C., & Tassa, Y. (2018). Safe Exploration in Continuous Action Spaces. ArXiv Preprint ArXiv:1801.08757.

[3] Chow, Y., Nachum, O., Faust, A., Duenez-Guzman, E., & Ghavamzadeh, M. (2019). Lyapunov-based Safe Policy Optimization for Continuous Control. Reinforcement Learning for Real Life (RL4RealLife) Workshop in the 36 Th International Conference on Machine Learning.

**Reviewer Expertise:**

Very good: Comprehensive knowledge of the area

**Strengths And Weaknesses:**

Strengths
1. This reformulation of the constraints considers the forward invariance and the finite-time convergence. This reformulation is beneficial for real robot applications.
2. The proposed method can be applied to other RL algorithms.

Weaknesses
1. Applying the proposed action search method AdamBA in high dimensional action space is challenging. You will need more samples to find a feasible direction or even not able to find one. Besides, the feasible action could highly deviate from the action sampled from the policy. This deviation could potentially destroy the learning process of the value function. In addition, it’s not possible to ensure that the optimized actions are the same at each time .
2. Improper baselines. The proposed method focuses on the safe exploration problem. But the author chose safe policy search methods as baselines. For comparison, safe exploration methods, e.g., Safety Layer[2] and SDDPG/SPPO[3], should be better choices.
3. The paper requires a forward dynamics model for the sample-based optimization. The claim of “model-free” is wrong.
4. The experiment only runs in the Safety Gym environment. It would be better to have some simulations with physical robots.
5. Each experiment has only three runs which can be highly biased. My suggestion is to run 25 random seeds to justify the claim for the RL problem.



**Summary Of Recommendation:**

The paper builds upon the work [1] and proposes a rule to choose the hyperparameter. The consideration of the forward invariance and the finite-time convergence is essential for robotics. The Safety Index Rule is determined based on the known dynamics model of the 2d differential drive robots. However, the proposed method is not model-free. In addition, the author did not choose proper baselines in comparison. And three runs for RL experiments are not sufficient to prove the performance.

---

> ### Author Response · Authors · 2021-08-28
> **Author Reply to Reviewer nDua (1/2)**
>
> We sincerely thank you for your comprehensive comments on our paper and please find our answers to your questions below.
> ___
> **Q1**: “Applying the proposed action search method AdamBA in high dimensional action space is challenging...”
>
> **A1**: To demonstrate the scalability of ISSA, we have added more experiments on ISSA with a complex 12-dimensional doggo robot. We added a new section (Section H) in the revised paper analyzing the scalability of AdamBA and ISSA in higher dimensional control systems. Please refer to A1 in Common Response and Section H in the revised paper for more detailed discussions. Note that for every unit gradient vector, the process of AdamBA outreach/decay is independent of the other unit gradient vectors, which means we could utilize parallel computation in practice when applying real-time robots. To demonstrate the scalability of AdamBA, We have tested the computation cost of parallel AdamBA and non-parallel AdamBA, where the computation time of parallel AdamBA (Figure18) remains nearly the same as the number of unit vectors exponentially increases, which shows the potential capability of AdamBA scaling to higher dimensional real-time control systems.
> ___
> **Q2**: “...the feasible action could highly deviate from the action sampled from the policy. This deviation could potentially destroy the learning process of the value function. In addition, it’s not possible to ensure that the optimized actions are the same at each time”
>
> **A2**: In Table 1, we have shown the number of vectors is a trade-off between performance (reward) and efficiency (computation time). The larger the number of vectors is, the more safe control points we could have as candidates to solve Equation 3, which leads to better reward performance (note that each safe control candidate on the safe set boundary is the suboptimal solution of equation 3). It is actually possible to ensure that the optimized actions are the same at each time, where we can set the random seed of AdamBA every time we enter it. Even if we can not ensure that the optimized actions of ISSA are the same, we could view the safety layer as a part of the stochastic environment, which is a common setting in RL formulation.
> ___
> **Q3**: “Improper baselines….”
>
> **A3**: We have added Safety Layer[2] as a baseline across all experiments as PPO-SL in our paper. Though Safety Layer[2] is a hard safe RL algorithm, it fails to decrease violation of cost in all environments as reported in Figure 5. The reason behind this is the fact that the linear approximation of cost function [1] has large approximation errors when the dynamics are highly nonlinear like the ones we used in MuJoCo. More importantly, PPO-SL cannot guarantee that these always exist a feasible safe control to lower the cost, since they directly use the user defined cost function which cannot always ensure feasibility [1]. We do highlight that our safety index design rule is able to guarantee a non-empty set of safe control for all states for 2-D plane systems, and ISSA is able to find safety control and achieve zero-violation with theoretical guarantee (Theorem 1).
>
> SDDPG/SPPO[3] also leverage the linear approximation of contriant in Safety Layer[2], which suffers the same problem as Safety Layer[2]. In the SDDPG/SPPO[3] original paper, SDDPG/SPPO[3] still violates the constraint during exploration like PPO-Lagrangian and CPO do. Since our paper aims at achieving zero-violation for RL algorithm, adding one more soft safe RL algorithm as baseline may not be necessary. So we have added SDDPG/SPPO[3] in the section of literature review of safe RL algorithms.
>
>
> [2] Dalal, G., Dvijotham, K., Vecerik, M., Hester, T., Paduraru, C., & Tassa, Y. (2018). Safe Exploration in Continuous Action Spaces. ArXiv Preprint ArXiv:1801.08757.
>
> [3] Chow, Y., Nachum, O., Faust, A., Duenez-Guzman, E., & Ghavamzadeh, M. (2019). Lyapunov-based Safe Policy Optimization for Continuous Control. Reinforcement Learning for Real Life (RL4RealLife) Workshop in the 36 Th International Conference on Machine Learning.

---

> ### Author Response · Authors · 2021-08-28
> **Author Reply to Reviewer nDua (2/2)**
>
> **Q4**: “The paper requires a forward dynamics model for the sample-based optimization. The claim of “model-free” is wrong. ”
>
> **A4**: We emphasize that “model-free” is to contrast against model-based safe control. Here ‘model-free’ refers to no explicit model being needed when synthesizing the safety index and searching for safe control. In contrast to previous methods requiring explicit knowledge about dynamics function, our proposed method does not require the explicit form of system dynamics, and our method can be guaranteed to generate safe control ​​as long as the safe set is regular (i.e., has non empty interior, almost everywhere differentiable boundary, etc. )
>
> ___
>
> **Q5**: “The experiment only runs in the Safety Gym environment. It would be better to have some simulations with physical robots.”
>
> **A5**: Due to Covid-19, we are not able to perform evaluation of ISSA on real physics robots. However, safetyGym is a well adopted benchmark and MuJoCo is an advanced physics simulator.
>
> ___
>
> **Q6**: “Each experiment has only three runs which can be highly biased.”
>
> **A6**: Due to the limited time of rebuttal and our computation power, we have added as many random seeds (Each experiment has five runs now) as we can for each experiment reported (in Figure 5, , Figure 8, Figure 15, Figure18 and Table 5). Five random seeds of experiment are commonly accepted in the RL community (Safety Gym benchmark itself only uses 3 seeds). And we will add more random seed runs to reduce bias in the camera-ready version.
>
> ___
>
> **Q7**: “There is no Figure 10a in the appendix.”
>
> **A7**: Thanks! We have corrected the  reference error in the revised paper.
>
> ___
>
> **Q8**: “Need more explanation about the safety index rule. It is not clear how the rule is determined.”
>
> **A8**: The full derivation of the safety index rule is in Appendix D.
>
> ___
>
> **Q9**: “...’ It can be shown that the ISSA algorithm is guaranteed to find a feasible solution …’. How?“
>
> **A9**: The proof of feasibility of ISSA to find a safe control is in Appendix E.

---

> ### Comment · Reviewer_nDua · 2021-08-31
> **Reply to the Author's Comment**
>
> Thank you for the detailed reply. The author's explanation and additional experiments have addressed my major concerns. I decide to change my decision to **weak accept**.
>
> However, the proposed Safety Index Rule can be more powerful than simply collision avoidance between two points. It would be beneficial if the authors could discuss more complex tasks, such as the manipulation task of avoiding collisions with obstacles (with geometric shapes) in the simulation.
>
> Additionally, in the collision avoidance task, the constraint is defined as $\phi=\sigma+d^n_{min} - d^n - k\dot{d}$. In order to get $\dot{d}$,  the velocity of the obstacles is known. It would be better if you can point the assumption out, in case you have any.
>
> Furthermore, I would suggest the author run 25 seeds for the learning task. Even many works have only 5 runs for their algorithms, it is not statistically sufficient for complex tasks. This paper (https://arxiv.org/pdf/1709.06560.pdf) shows that two same experiment setups with 5 different seeds result in almost zero-overlapping learning curves.

---

### Official Review · Reviewer_2HkF · 2021-07-21

**Originality:** Good
**Technical Quality:** Very Good
**Clarity Of Presentation:** Good
**Impact:** 4

**Recommendation:**

Weak Accept: I recommend accepting the paper, but will not argue for my recommendation if the majority of other reviewers have a different opinion.

**Summary:**

This paper proposes the model-free Implicit Safe Set Algorithm (ISSA), a framework for zero-violation control that uses energy function-based methods to ensure robot safety (represented in this paper by collision avoidance) when only a black-box dynamics simulator is available, rather than an analytical model. ISSA is inspired by the model-based safe set algorithm of Liu and Tomizuka (2014), developed with the goal of extending their algorithm's capabilities to systems where only a black-box simulator is available. To formulate ISSA, a design rule is proposed to synthesize the energy function (termed the safety index in this paper) so that a feasible control is available at any state, and a sample-efficient algorithm for constrained black-box optimization is proposed for sampling the control space to find a control that dissipates energy and maximizes reward. ISSA is evaluated over several Safety Gym benchmark problems, serving as a safety layer used in conjunction with the PPO reinforcement learning algorithm. The resulting PPO-ISSA framework, in exchange for a modest reduction in rewards, avoids collision entirely while competing RL algorithms do not.





**Issues:**


-The safety index design rule used in the paper is specific to collision avoidance by a mobile robot. It would be helpful if the authors could clarify the intended pathway toward more general, systematic formulation of appropriate design rules for the various systems of interest for safe RL.

-Table 1 is helpful in addressing the dependence of ISSA's computational expense on the number of vectors, but the normalization of the required computation time also obfuscates ISSA's compatibility with real-time robotics applications. It would be helpful if the authors could comment further on the computational expense of using an ISSA safety layer, and the computing arrangements that would be needed to support its use for safe real-time control in the course of learning with real robotic systems.

-Here are a few grammatical and typographical errors that I noticed while reading the paper:
--Line 95: "methods is able" should be "methods are able"
--Line 107: "methods still exploits" should be "methods still exploit"
--Line 119: "property of set" should be "properties of the set"
--To my knowledge, safeguard is typically spelled as one word, rather than "safe guard" as used throughout the paper


**Reviewer Expertise:**

Good: General knowledge of the area

**Strengths And Weaknesses:**


Strengths:

-The paper proposes a practical approach for achieving a well-motivated and highly desirable capability - avoiding collision entirely in the course of learning, in the absence of an analytical dynamics model for the system of interest.

-The paper is well-organized and includes helpful illustrative figures to explain the architecture and working principles of the proposed algorithm, the settings in which it is evaluated, and the results of the experiments performed.

-The authors establish proof of the beneficial properties of forward invariance and finite-time convergence for the proposed algorithm.

-The authors have made their source code freely available as a supplement to their paper.

Weaknesses:

-The paper describes a framework for safe control that can be highly useful for learning, but is not a learning method at its core.

-The safety index design rule adopted in the paper is specific to the problem of collision avoidance for a mobile robot.




**Summary Of Recommendation:**

I believe the algorithm proposed and evaluated in this paper, intended for use as a safety layer in a reinforcement learning framework, constitutes a novel and highly useful capability when safety is a primary concern. Although the proposed framework is composed of a collection of arguably incremental advances, there seems to be great potential for impact and adoption by others in safe RL applications.

---

> ### Author Response · Authors · 2021-08-28
> **Author Reply to Reviewer 2HkF**
>
> We sincerely thank you for your comprehensive comments on our paper and please find our answers to your questions below.
>
> ___
> **Q1**: “The paper describes a framework for safe control that can be highly useful for learning, but is not a learning method at its core.”
>
> **A1**: Previous RL methods have been criticized due to the lack of probable safety. In this paper, we aim to address the critical problem of providing probable safety for learning methods, and the proposed method can be applied to any other RL algorithms. We believe our proposed methods are beneficial towards the real-world application of robot learning.
>
> ___
>
> **Q2**: “The safety index design rule used in the paper is specific to collision avoidance by a mobile robot.”
>
> **A2**: In this paper, we are specifically interested in the safety of collision avoidance in 2-D plane. The general safety index synthesis rule for other safety-critical situations is summarized in Equation 8 and Equation 14 in Appendix D.1.
>
> ___
>
> **Q3**: “Table 1 is helpful in addressing the dependence of ISSA's computational expense on the number of vectors, but the normalization of the required computation time also obfuscates ISSA's compatibility with real-time robotics applications. It would be helpful if the authors could comment further on the computational expense of using an ISSA safety layer, and the computing arrangements that would be needed to support its use for safe real-time control in the course of learning with real robotic systems.”
>
> **A3**: In the revised paper, we have reported the unnormalized computation time of ISSA (in Table 6) under different choices of number of unit vectors and robots. More importantly, for every unit gradient vector, the process of AdamBA outreach/decay is independent of the other unit gradient vectors, which means we could utilize parallel computation in practice when applying real-time robots. We have tested the computation cost of parallel AdamBA and non-parallel AdamBA, where the computation time of parallel AdamBA remains nearly the same as the number of unit vectors exponentially increases. Overall, we have shown the capability of AdamBA and ISSA scaling to higher dimensional real-time control systems. Please refer to A1 in Common Response and Section H in the revised paper for more detailed discussions.
>
> ___
>
> **Q4**: “...a few grammatical and typographical errors...”
>
> **A5**: Thanks! We have corrected the grammatical and typographical errors.

---

> > ### Comment · Reviewer_2HkF · 2021-09-03
> > **Post-rebuttal comments**
> >
> > This comment is to acknowledge that I have read the authors' revisions and responses to the reviewers. I appreciate the authors' efforts to address the reviewers' concerns, although my original rating still best-represents my views on the paper and its suitability for inclusion in the CoRL proceedings.

---

### Official Review · Reviewer_yCZE · 2021-07-23

**Originality:** Good
**Technical Quality:** Very Good
**Clarity Of Presentation:** Very Good
**Impact:** 2

**Recommendation:**

Weak Accept: I recommend accepting the paper, but will not argue for my recommendation if the majority of other reviewers have a different opinion.

**Summary:**

This paper presents a method for safe reinforcement learning based on the notion energy-based methods to ensure invariance of certain properties, or sets. Importantly for the context of modern RL, the method is model free and comes with the capability to reason about black box or unknown dynamics and provide a kind of barrier certificate at training time. The paper is well written and the ideas are sound. I have a few intellectual questions below, but the large part of my comments are editorial.

**Issues:**

This is not a huge issue, but on lines 70 and 71 the authors use the word "implicit", which I believe I understand from the context; however, "implicit" is used in the related context of model-predictive control to mean something different. If the authors can think of a synonym, this might be helpful, e.g. "unknown"???

In section 2, is it trivial for the user to specify phi_0 for the safety specification? In practice, I believe this can be quite challenging - I recognize this is likely out of scope for the contributions of the paper but simply note this is a potential hurdle to adoption.

Line 77: "subjected to be modified" reads a bit clunky. I suggest: "subject to modification by the safe guard".

Line 87: why would X_s contain states that will inevitably go to an unsafe state? Why not simply define an invariant set and just call that the "safe set"? It is a bit strange to call something a safe set when it will inevitably lead to an unsafe state - perhaps X_S should be called something else? Just a generic 0-sublevel set and remove the word "safe"? Sorry, this is just a bit strange to me, although I understand the technical rationale for needing a subset.

Line 95: These methods *are* able to... (typo)

Minor but general comment: the paper is well-written, including very strong prose. However, there are several typos, including things like subject-verb agreement, etc. Please clean this up in the final (again, not a huge deal but worth a bit of extra effort, grammerly, etc)

Line 119: perhaps we will get to this later in the paper, but the authors suggest that safe control is guaranteed, regardless of the geometry of the safe set. Can the set be disjoint or unconnected?

In lines 120-143, the authors provide a sketch of safe control, focusing on safe set algorithms. Utlimately, most of the detail is in Appendix A, which is fine to some extent, but for "self contained-ness", the paper would be improved by helping the reader a bit more here. I am struggling to come up with a helpful suggestion here, but hopefully this helps: instead of vaguely discussing things like adding higher-order terms, the discourse could be made more precise by directly including the details from appendix A (i.e. equation 5 and the subsequent text), as well as the discussion in appendix A about what differs between this paper and the literature. I recognize that there are space constraints, but I believe this would improve readability

General question: is it correct to say that SSA is a generalization of control barrier functions?

Note that in line 149-150, the authors essentially do what I had asked earlier about the term "implicit". Perhaps this statement could be used earlier, although this is again not a huge deal

Equation 3 brings to bear the fact that (I think) obstacles are assumed to be static. I might have missed this assumption in the intro or elsewhere, but it is worth making this assumption clear

Is the approximation summarized in line 158, and described in Appendix B, guaranteed to be a conservative approximation? I do not believe that either the main body or App B actually makes this assertion and/or proof. Appendix G provides a numerical example but no proof of conservatism. I could be wrong here - if that is the case, please make this clear either in section 4 or Appendix B, or both

AdamBA seems to be "expensive", and is subject to the number of unit gradient vectros that are initialized, the dimensionality of the safe control set, etc. Can the authors comment on computational cost?

This is an extremely minor point (almost embarrasing to say this), but the gradient directions in figure 1(a) are slightly misaligned from those in (b)-(d), especially the 3 vectors that are pointed "down". It seems like an easy fix and is worth it, in my opinion, to clarify the AdamBA method

The authors state that the reward performance of PPO-ISSA is comparable to the other methods, but I am not sure this claim is justified. The method constitently has lower performance than the other methods. It might be better to couch this argument in terms of the "cost" of a safety violation and emphasize that there is a tradeoff here, and (depending on how performance and safety are traded off) ISSA does better

PPO is a reasonable starting place to provide a baseline. I have no problem with this, but I suggest the authors justify this choice, somewhere in the neighborhood of line 236. For example, one reason this is a good baseline is because (a) PPO is relatively state-of-the-art but (b) is generic and (c) already has safety-constrained derivatives that can be tested off the shelf. There is so much going on with RL that it is impossible to have a perfect baseline; again, I agree with the choice but feel that the text could provide a stronger justification.

The figures are in strange locations, which I understand: there are space limitations and typesetting is difficult. However, figure 6, for example, belongs closer to the body text that discusses it. On line 275, figure 7 is mentioned but the figure immediately to the right is figure 6. Perhaps after all other revisions are complete, the authors could attempt to re-optimize in terms of figure placement. Again, not a huge deal and this is not a technical comment, but a readability comment.

In the conclusion, the authors report 98% pm 22% reward performance relative to SotA methods. The "zero safety violation" claim is easy enough to see from the paper, but I am not sure where this 98% number comes from. It does not make sense based on figure 5, and I am unclear about the method used to derive these numbers. Did I miss it? Is it in the appendix? I believe the number is derived from section G.2.4, but this section seems to only include one scenario.

**Reviewer Expertise:**

Very good: Comprehensive knowledge of the area

**Strengths And Weaknesses:**

The main strengths of this paper include: organization, clarity of writing, comparison to related works and/or context of approach. The approach and its presentation in the paper is/are technically sound.

I do not have any major suggestions for improvement, other than that the significance of the contribution is unclear to me. I am not saying there is *not* a significant amount of originality or impact, but based on my knowledge of the topic, this paper feels somewhat incremental.

**Summary Of Recommendation:**

Safety in robotics applications, particularly vis a vis reinforcement learning and black box models, is an important and active area. This is a sound contribution. Assuming that some of my comments are addressed, I believe this paper would represent a nice contribution to the conference.

---

> ### Author Response · Authors · 2021-08-28
> **Author Reply to Reviewer yCZE (1/2)**
>
> We sincerely thank you for your comprehensive comments on our paper and please find our answers to your questions below.
> ___
> **Q1**: “...on lines 70 and 71 the authors use the word ‘implicit’...” “Perhaps this statement could be used earlier...”
>
> **A1**: We have added the definition of “implicit” to the problem formulation in line 72 where the word “implicit” first appears.
>
> ___
>
> **Q2**: “...In section 2, is it trivial for the user to specify phi_0 for the safety specification?...”
>
> **A2**: It should be straightforward for the user to specify phi_0 by directly using the most intuitive specification. For example, for collision avoidance safety specification, where $\phi_0$ can be designed as the negative closest distance between the robot and environmental hazards.
>
> ___
>
> **Q3**: “‘subjected to be modified’ reads a bit clunky. I suggest: ‘subject to modification by the safe guard’.”
>
> **A3**: Thanks! We have revised the statement as you suggested.
>
>
> ___
> **Q4**: “why would X_s contain states that will inevitably go to an unsafe state? … perhaps X_S should be called something else?”
>
> **A4**: We have clarified in the revised paper that $X_s$ as the user-defined safe set. And say all states in X_s are safe, but it is not guaranteed they will remain safe.
>
>
> ___
> **Q5**: “...there are several typos...”
>
> **A5**: Thanks for pointing this out! We have corrected all the typos we found and checked with grammarly.
>
>
> ___
> **Q6**: “...the authors suggest that safe control is guaranteed, regardless of the geometry of the safe set. Can the set be disjoint or unconnected?”
>
> **A6**: The safe set should be a zero sublevel set of a continuous function which is differentiable almost everywhere. It can be disjoint and unconnected.
>
> ___
> **Q7**: “...but for "self contained-ness", the paper would be improved by helping the reader a bit more here….”
>
> **A7**: Thanks for this suggestion! To improve the self-contained-ness of our paper, we have made the discourse more precise (line 116 - line 123)  by directly including the details from appendix A of previous paper, and added how our paper differs from the safe control literature in the main text in Section 3 (line 127 - line 134).
>
>
> ___
> **Q8**: “General question: is it correct to say that SSA is a generalization of control barrier functions?”
>
> **A8**: SSA was introduced in parallel with CBF and a detailed comparison is included in [1]. Roughly, we can say SSA is a generalization of CBF.
>
>
> ___
> **Q9**: “Equation 3 brings to bear the fact that (I think) obstacles are assumed to be static. I might have missed this assumption in the intro or elsewhere, but it is worth making this assumption clear”
>
> **A9**: We have updated our safety index synthesis rule while explicitly pointing out the assumption we rely on, and we also updated the theoretical proofs of proposition 1 in the appendix D.2. We also explicitly stated that the obstacles can be static or non-static in line 152 - 153.
>
>
> ___
> **Q10**: “Is the approximation summarized in line 158, and described in Appendix B, guaranteed to be a conservative approximation? I do not believe that either the main body or App B actually makes this assertion and/or proof. Appendix G provides a numerical example but no proof of conservatism. I could be wrong here - if that is the case, please make this clear either in section 4 or Appendix B, or both”
>
> **A10**: This is a very interesting part, thanks for pointing this out! We apologize that the previous statement of approximation from Appendix B might cause some confusion. In the updated version, we only preserved (6) and the text before (6) to demonstrate why (2) cannot be used in a discrete case when dt is not sufficiently small, and then we point out the more proper constraint (described in line 148) to guarantee safety in a discrete time systems. We then gave a proof to explicitly demonstrate that by satisfying the constraint in (3), we ensure finite time convergence and forward invariance in the discrete system. (line 418 - 422)

---

> ### Author Response · Authors · 2021-08-28
> **Author Reply to Reviewer yCZE (2/2)**
>
> **Q11**: “AdamBA seems to be "expensive", and is subject to the number of unit gradient vectors that are initialized, the dimensionality of the safe control set, etc. Can the authors comment on computational cost?”
>
> **A11**: To demonstrate the scalability of ISSA, we have added more experiments on ISSA with a complex 12-dimensional doggo robot. We added a new section (Section H) in the revised paper analyzing the scalability of AdamBA and ISSA in higher dimensional control systems. Please refer to A1 in Common Response and Section H in the revised paper for more detailed discussions. Note that the process of AdamBA outreach/decay for each unit gradient vector is independent from each other, thus we can always accelerate ISSA by parallel computation in practice to apply ISSA in real-time robotics application. To demonstrate the scalability of AdamBA to higher control dimensions, We have tested the computation cost of parallel AdamBA and non-parallel AdamBA, where the computation time of parallel AdamBA (Figure18) remains nearly the same while the number of unit vectors exponentially increases, which shows the potential capability of AdamBA scaling to higher dimensional real-time control systems.
>
> ___
>
>
> **Q12**: “the gradient directions in figure 1(a) are slightly misaligned from those in (b)-(d)”
>
> **A12**: Thanks! We have revised the figure.
>
> ___
>
> **Q13**: “The authors state that the reward performance of PPO-ISSA is comparable to the other methods … It might be better to couch this argument in terms of the "cost" of a safety violation and emphasize that there is a tradeoff here, and (depending on how performance and safety are traded off) ISSA does better”
>
> **A13**: We have revised the discussion in Section 6.1, emphasizing that PPO-ISSA is able to achieve zero average episode cost and zero cost rate across all experiments, in exchange with a slight sacrifice of the reward performance. And we emphasize the tradeoff in the metrics comparison section.
>
>
> ___
>
> **Q14**: “PPO is a reasonable starting place to provide a baseline ... I agree with the choice but feel that the text could provide a stronger justification. ”
>
> **A14**: We have revised our paper following your suggestions and added a stronger justification about the choice of PPO as the baseline RL algorithm.
>
>
> ___
>
> **Q15**: “The figures are in strange locations, which I understand: there are space limitations and typesetting is difficult.”
>
> **Q15**: We really appreciate your suggestions and understanding. After all other revisions are complete, we will attempt to re-optimize in terms of figure placement to improve readability of the camera-ready version.
>
>
> ___
>
> **Q16**: “the authors report 98% pm 22% reward performance relative to SotA methods. The "zero safety violation" claim is easy enough to see from the paper, but I am not sure where this 98% number comes from.”
>
> **A16**: We updated the description of reported performance results in the revised paper due to more random seeds and more baselines. The detailed calculation is now introduced in G.2.4, where we we calculate the converged reward $\bar{J}_r$ percentage of PPO-ISSA compared to other three safe RL baseline methods (PPO-Lagrangian, CPO and PPO-SL) over eight control suites. The computed mean reward percentage is $95\%$, and the standard deviation is $9 \%$. Therefore we conclude that PPO-ISSA is able to gain $95\% \pm 9\% $ cumulative reward compared to state-of-the-art safe DRL methods.
>
> ___
>
> [1]Wei, T., & Liu, C. (2019, December). Safe control algorithms using energy functions: A uni ed framework, benchmark, and new directions. In 2019 IEEE 58th Conference on Decision and Control (CDC) (pp. 238-243). IEEE.

---

### Official Review · Reviewer_VywJ · 2021-07-23

**Originality:** Fair
**Technical Quality:** Fair
**Clarity Of Presentation:** Good
**Impact:** 2

**Recommendation:**

Weak Accept: I recommend accepting the paper, but will not argue for my recommendation if the majority of other reviewers have a different opinion.

**Summary:**

In this paper the authors present a method for safe reinforcement learning where the dynamics of the system are unknown, i.e. black box. The crux of the problem is how to synthesize safe controls given that no explicit analytical form for the system can be used. To do so, the authors design a safety index, a Lyapunov-like function, which implicitly defines the set of safe actions. If the nominal action to be taken is outside the safe set, a new action is produced, which is the L2 projection of the nominal action onto the safe set. The problem is that for black-box optimization problems this cannot be done efficiently without querying. The authors recognize that the optimal solution will always be at the boundary of the safe set of controls, and so they design AdamBA to find near-optimal points using a 2 phase algorithm. The first phase generates units vectors which are extended exponentially until the control bounds are exceeded, or the safe set is found. If the safe set is found then binary search is used to find safe controls at the boundary. If no safe controls are found, an anchor control point is found by uniform sampling, and AdamBA is executed again. In the result section, the authors showcase the algorithm on a Point Robot system from Safety Gym and show how PPO-ISSA is able to learn competitive policies when compared to PPO, PPO-Lagrangian and CPO, without violating safety.

**Issues:**

As previously stated my main concern is with the scalability of the method. Most robotic systems have more than 2 control dimensions. To seriously consider this paper for acceptance I would expect a more realistic robotic system than a point robot.

**Reviewer Expertise:**

Good: General knowledge of the area

**Strengths And Weaknesses:**

Strengths:
* The paper addresses a crucial important problem in robotics, which is that of learning while remaining safe.
* The paper uses well-grounded concepts from control theory through the use of the safety index.

Weaknesses:
* It is not clear how well AdamBA can scale to higher dimensional control systems. As the control dimension increases I would expect the sampling strategies to fail or take too long.
* In the same spirit as the previous comment, the experimental section deals only with a single system with only a 2-dimensional control space.

**Summary Of Recommendation:**

I did enjoy reading the paper, but I am afraid that without some more solid analysis and evidence for the scalability of the method to systems with more control dimensions, I would not accept it in its current form.

---

> ### Author Response · Authors · 2021-08-28
> **Author Reply to Reviewer VywJ**
>
> We sincerely thank you for your comprehensive comments on our paper and please find our answers to your questions below.
> ___
> **Q1**: “It is not clear how well AdamBA can scale to higher dimensional control systems. As the control dimension increases I would expect the sampling strategies to fail or take too long.”
>
> **A1**:  Note that the process of AdamBA outreach/decay for each unit gradient vector is independent of each other, and thus we can always accelerate ISSA by parallel computation in practice to apply ISSA for real-time robot applications. We have tested the computation cost of parallel AdamBA and non-parallel AdamBA, where the computation time of parallel AdamBA remains nearly the same regardless of the exponentially increased number of unit vectors, which shows the potential capability of AdamBA scaling to higher dimensional real-time control systems.
>
>
> ___
> **Q2**: “The experimental Appendix Deals only with a single system with only a 2-dimensional control space.”
>
> **A2**: To demonstrate the scalability of ISSA, we added a new section (Section H) in the revised paper analyzing the scalability of AdamBA and ISSA in higher dimensional control systems. We have tested ISSA with a 12-dimensional doggo robot. The experiment results (Figure 8) show that ISSA is able to guarantee zero-violation and achieve higher reward compared to baselines. And we only need to use twice more vectors in ISSA phase-1 for a 12-dimensional doggo robot compared to a 2-dimensional point robot. We have reported the average computation time of ISSA under different choices of robots and number of vectors (in Table 6), which does not show a significant increase of the computation time when the dimensionality of the robot states and controls increase. . In fact, as long as the safety index design is properly designed (i.e., that there is always a large feasible safe control set at every state), ISSA only needs to generate a small number of search directions for a safe control, which mitigates the computation cost for ISSA. Note that we can increase the size of feasible  safe control  in the control space with bigger $k$ as shown in Figure 7. In this way, we can decrease the number of vectors needed to search for a safe control. Please refer to A1 in Common Response and Section H in the revised paper for more detailed discussions.
>
>
> ___
> **More**:
> Apart from the proposed algorithm AdamBA and ISSA to find near-optimal safe controls. Another major contribution of our paper is that we propose a safety index design rule for collision avoidance of 2-D plane. And most importantly, with the safety index design rule and ISSA, we give theoretical proof of forward invariance safety. Provable safety in theory is prominently weighted in the reward-safety tradeoff, since any safety violation may lead to property loss, life danger in the real robotics applications.

---

> ### Comment · Reviewer_VywJ · 2021-08-31
> **Thanks for the updates**
>
> I would like to thank the authors for updating the paper. I have change my decision based on the doggo results.

---

### Author Response · Authors · 2021-08-28
**Common Response to All Reviewers**

We sincerely thank all reviewers for all the detailed and helpful reviews!

We have revised the paper to address all the important issues raised in the reviews. The revision part in the updated PDF is highlighted in red.

For your convenience, we provide the full revised paper (MainText + Appendix) in the supplementary materials.

Among all the reviews, there are some common questions about our paper we would like to answer here.

___
**Q1**: How well ISSA can scale to higher dimensional control systems?

**A1**: We added a new section (Section H) in the revised paper analyzing the scalability of AdamBA and ISSA in higher dimensional control systems. We have tested ISSA with a 12-dimensional doggo robot. The experiment results (Figure 8) show that ISSA is able to guarantee zero-violation and achieve higher reward compared to baselines. And we only need to use twice more vectors in ISSA phase-1 for a 12-dimensional doggo robot compared to a 2-dimensional point robot. We have reported the average computation time of ISSA under different choices of robots and number of vectors (in Table 6), which does not show a significant increase of the computation time when the dimensionality of the robot states and controls increase. In fact, as long as the safety index design is properly designed (i.e., that there is always a large feasible safe control set at every state), ISSA only needs to generate a small number of search directions for a safe control, which mitigates the computation cost for ISSA. Note that we can increase the size of feasible  safe control  in the control space with bigger $k$ as shown in Figure 7. In this way, we can decrease the number of vectors needed to search for a safe control.

More importantly, we also highlight the fact that the process of AdamBA outreach/decay for each unit gradient vector is independent of each other, and thus we can always accelerate ISSA by parallel computation in practice to apply ISSA for real-time robot applications. We have tested the computation cost of parallel AdamBA and non-parallel AdamBA, where the computation time of parallel AdamBA remains nearly the same regardless of the exponentially increased number of unit vectors. Overall, we have shown the capability of AdamBA and ISSA scaling to higher dimensional real-time control systems.

___
**Q2**: “...The safety index design rule only applies to mobile robot collision avoidance and seems to implicitly require obstacles to be static, something that is not stated in the text….”

**A2**: In this paper, we are specifically interested in the safety of collision avoidance for 2-D plane. The general safety index synthesis rule for other safety-critical situations is summarized in Equation 8 and Equation 14 in Appendix D.1. Besides, we have added more descriptions (in Section 4.1) to clarify the assumptions/conditions about our safety index design rule. Note that we also updated the proof of Proposition 1 in Appendix D.2, which doesn’t require the obstacles to be static. Thus our safety index design rule can be both applied to static and non-static obstacles.

---

### Meta-Review · Area_Chair_fZ9n · 2021-08-14

**Recommendation:** Accept (Poster)
**Confidence:** 5

**Metareview:**

### Final Meta-Review

The authors have carefully responded to the questions and concerns raised in the reviews and meta-review; they have made improvements to the clarity and rigor of the exposition in response to issues brought up by reviewers; and they have included additional simulation results for a high-dimensional nonlinear system (quadrupedal robot) to provide empirical evidence of the method's scalability.

These changes have significantly strengthened the final manuscript and, together with the thoughtful clarifications offered during the discussion, have brought all reviewers to an agreement on a positive recommendation for the paper, which I am happy to endorse.

### Original Meta-Review

The paper proposes a safe reinforcement learning approach that uses a black-box dynamics function to ensure safety. The reviewers agree on the importance of the topic and overall find the paper to be clearly written. However, the reviewers bring up a number of concerns, primarily regarding the exposition and results.

There appear to be a number of unstated assumptions made throughout the paper that detract form the claimed generality of the approach. Reviewers yCZE and 2HkF point out that the safety index design rule (3) only applies to mobile robot collision avoidance and seems to implicitly require obstacles to be static, something that is not stated in the text. There is also reviewer concern that the exposition of the proposed approach is not sufficiently self-contained, with some critical information relegated to the Appendix.

Evaluation results are limited to a comparatively simple dynamical system; reviewers Vyw and nDua indicate likely scalability challenges with the control space dimension, which are not addressed in the paper. There are further reviewer concerns that the number of simulation runs is insufficient for statistical significance, and that the chosen baselines may not provide an appropriate comparison because they are not fully safe learning methods, which could lead to the incorrect perception that the proposed method is the first to enable zero-violation learning.

Overall, the authors must respond to the detailed issues raised by the reviewers around the paper's claims, technical exposition, and experimental evaluation.

In addition, I ask that the authors clarify the following issue that I encountered while preparing this meta-review: given that there are no stated assumptions on the system dynamics f(x,u), the Safety Index Design Rule (3) is implied to be valid for extremely general cases, and the results in Proposition 1 and Theorem 1 therefore seem quite strong as stated. However, in the Appendix it is revealed that both of these results rely on the dynamics corresponding to a differential drive robot. This seems very restrictive (especially for an assumption that only appears in the Appendix), and appears to violate a central claim of the paper: if constructing a safety index requires knowledge of the specific form of the dynamics f(x,u), this would mean that that this is not in fact a black-box-compatible method.

---

> ### Author Response · Authors · 2021-08-28
> **Author Reply to Area Chair fZ9n (1/2)**
>
>
> We sincerely thank you for your comprehensive comments on our paper and please find our answers to your questions below.
> ___
>
> **Q1**: “There appear to be a number of unstated assumptions made throughout the paper that detract from the claimed generality of the approach.”  “...given that there are no stated assumptions on the system dynamics f(x,u), the Safety Index Design Rule is implied to be valid for extremely general cases...”
>
> **A1**: We have added more descriptions (in Section 4.1) to clarify the assumptions of the safety index design rule. The safety index design rule is derived for the collision avoidance problem for 2-D plane systems.
>
> ___
>
> **Q2**: “...the safety index design rule only applies to mobile robot collision avoidance and seems to implicitly require obstacles to be static...”
>
> **A2**: In this paper, we are specifically interested in the safety of collision avoidance for 2-D plane. The general safety index synthesis rule for other safety-critical situations is summarized in Equation 8 and Equation 14 in Appendix D.1, showing the intended pathway towards more general, systematic formulation of appropriate design rules for the various systems of interest for safe RL. We also updated the proof of Proposition 1 in Appendix D.2, which doesn’t require the obstacles to be static. Thus our safety index design rule can be both applied to static and non-static obstacles.
>
> ___
>
>
> **Q3**: “...the exposition of the proposed approach is not sufficiently self-contained, with some critical information relegated to the Appendix.”
>
> **A3**:  To improve the self-contained-ness of our paper. We have made the discourse more precise (line 116 - line 123)  by directly including the details from appendix A of previous paper, and added how our paper differs from the safe control literature in the main text in Section 3 (line 127 - line 134).
>
> ___
>
> **Q4**: “Evaluation results are limited to a comparatively simple dynamical system”
>
> **A4**: Due to Covid-19, we are not able to perform evaluation of ISSA on real physics robots. However, safetyGym is a well adopted benchmark and MuJoCo is an advanced physics simulator. To demonstrate the scalability of ISSA, we have added more experiments on ISSA with a complex 12-dimensional doggo robot. We added a new section (Section H) in the revised paper analyzing the scalability of AdamBA and ISSA in higher dimensional control systems. Please refer to A1 in Common Response and Section H in the revised paper for more detailed discussions.
>
> ___
>
> **Q5**: “...concerns that the number of simulation runs is insufficient for statistical significance...”
>
> **A5**: Due to the limited time of rebuttal and our computation power, we have added as many random seeds (Each experiment has five runs now) as we can for each experiment reported (in Figure 5, Figure 8, Figure 15, Figure 17 and Table 5). Five random seeds of experiment are commonly accepted in the RL community (Safety Gym benchmark itself only uses 3 seeds). And we will add more random seed runs to reduce bias in the camera-ready version.

---

> ### Author Response · Authors · 2021-08-28
> **Author Reply to Area Chair fZ9n (2/2)**
>
> **Q6.1**: “...the chosen baselines may not provide an appropriate comparison because they are not fully safe learning methods...”
>
> **A6.1**:  We have added the fully safe learning methods Safety Layer[1] as a baseline PPO-SL (reported in Figure 5, Figure 8, Figure 15, Figure 17, and Table 5) and added discussion about Safety Layer[1] in Section 6.1. Though Safety Layer[1] achieves zero-violation in its original paper, it fails to decrease violation of cost in all environments as reported in Figure 5. The reason behind this is the fact that the linear approximation of cost function [1] has large approximation errors when the dynamics are highly nonlinear like the ones we used in MuJoCo. More importantly, PPO-SL cannot guarantee that these always exist a feasible safe control to lower the cost, since they directly use the user defined cost function which cannot always ensure feasibility [1]. We do highlight that our safety index design rule is able to guarantee a non-empty set of safe control for all states for 2-D plane systems, and ISSA is able to find safety control and achieve zero-violation with theoretical guarantee (Theorem 1).
>
> **Q6.2**: “... which could lead to the incorrect perception that the proposed method is the first to enable zero-violation learning...”
>
> **A6.2**: Previous RL methods that enable zero-violation learning like Safety Layer[1] and ShieldNN[2] are limited to simple control-affine systems or some specific systems like kinematic bicycle models (KBM). Other zero-violation learning methods [3][4] still exploit the knowledge of the explicit form of system dynamics to guarantee constraint satisfaction. In contrast, our proposed method does not require the explicit form of system dynamics.
>
>
> ___
>
> **Q7**: “...if constructing a safety index requires knowledge of the specific form of the dynamics f(x,u), this would mean that this is not in fact a black-box-compatible method.”
>
> **Q7**: We are not addressing an extremely general case as answered above, the rule is applied to collision avoidance of 2-D plane while the assumption is explicitly updated in 4.1, and the general rule is mentioned in G.1. We don’t need the dynamics f(x,u), in fact, any point mass will have its maximum relative velocity to the obstacle, and maximum relative acceleration to obstacle. This is unrelated to dynamics but rather a system property, which can be measured without knowing the exact system dynamics. (For example, safety index rule can apply to single integrator, double integrator or unicycle without knowing their explicit dynamics)
>
> ___
> [1] Dalal, G., Dvijotham, K., Vecerik, M., Hester, T., Paduraru, C., & Tassa, Y. (2018). Safe Exploration in Continuous Action Spaces. ArXiv Preprint ArXiv:1801.08757.
>
> [2] J. Ferlez, M. Elnaggar, Y. Shoukry, and C. Fleming.  Shieldnn:  A provably safe nn filter for unsafe nn controllers.arXiv preprint arXiv:2006.09564, 2020.
>
> [3]J. F. Fisac, A. K. Akametalu, M. N. Zeilinger, S. Kaynama, J. Gillula, and C. J. Tomlin.  A general safety framework for learning-based control in uncertain robotic systems.IEEE Transactions on Automatic Control, 64(7):2737–2752, 2018.
>
> [4]F. Berkenkamp, M. Turchetta, A. P. Schoellig, and A. Krause. Safe model-based reinforcement learning with stability guarantees.arXiv preprint arXiv:1705.08551, 2017

---

### Decision · Program_Chairs · 2021-09-13

**Decision:**

Accept (Poster)

**Comment:**

### Final Meta-Review

The authors have carefully responded to the questions and concerns raised in the reviews and meta-review; they have made improvements to the clarity and rigor of the exposition in response to issues brought up by reviewers; and they have included additional simulation results for a high-dimensional nonlinear system (quadrupedal robot) to provide empirical evidence of the method's scalability.

These changes have significantly strengthened the final manuscript and, together with the thoughtful clarifications offered during the discussion, have brought all reviewers to an agreement on a positive recommendation for the paper, which I am happy to endorse.

### Original Meta-Review

The paper proposes a safe reinforcement learning approach that uses a black-box dynamics function to ensure safety. The reviewers agree on the importance of the topic and overall find the paper to be clearly written. However, the reviewers bring up a number of concerns, primarily regarding the exposition and results.

There appear to be a number of unstated assumptions made throughout the paper that detract form the claimed generality of the approach. Reviewers yCZE and 2HkF point out that the safety index design rule (3) only applies to mobile robot collision avoidance and seems to implicitly require obstacles to be static, something that is not stated in the text. There is also reviewer concern that the exposition of the proposed approach is not sufficiently self-contained, with some critical information relegated to the Appendix.

Evaluation results are limited to a comparatively simple dynamical system; reviewers Vyw and nDua indicate likely scalability challenges with the control space dimension, which are not addressed in the paper. There are further reviewer concerns that the number of simulation runs is insufficient for statistical significance, and that the chosen baselines may not provide an appropriate comparison because they are not fully safe learning methods, which could lead to the incorrect perception that the proposed method is the first to enable zero-violation learning.

Overall, the authors must respond to the detailed issues raised by the reviewers around the paper's claims, technical exposition, and experimental evaluation.

In addition, I ask that the authors clarify the following issue that I encountered while preparing this meta-review: given that there are no stated assumptions on the system dynamics f(x,u), the Safety Index Design Rule (3) is implied to be valid for extremely general cases, and the results in Proposition 1 and Theorem 1 therefore seem quite strong as stated. However, in the Appendix it is revealed that both of these results rely on the dynamics corresponding to a differential drive robot. This seems very restrictive (especially for an assumption that only appears in the Appendix), and appears to violate a central claim of the paper: if constructing a safety index requires knowledge of the specific form of the dynamics f(x,u), this would mean that that this is not in fact a black-box-compatible method.